# Melatonin Mediates Enhancement of Stress Tolerance in Plants

**DOI:** 10.3390/ijms20051040

**Published:** 2019-02-27

**Authors:** Biswojit Debnath, Waqar Islam, Min Li, Yueting Sun, Xiaocao Lu, Sangeeta Mitra, Mubasher Hussain, Shuang Liu, Dongliang Qiu

**Affiliations:** 1College of Horticulture, Fujian Agriculture and Forestry University, Fuzhou, Fujian 350002, China; biswo26765@yahoo.com (B.D.); liminzyl@sina.com (M.L.); yuetingsun@126.com (Y.S.); xc531599541@126.com (X.L.); sangeeta.dae@hotmail.com (S.M.); mubasherhussain05uaf@yahoo.com (M.H.); liushuangsyau@aliyun.com (S.L.); 2Department of Horticulture, Sylhet Agricultural University, Sylhet 3100, Bangladesh; 3College of Geographical Sciences, Fujian Normal University, Fuzhou, Fujian 350007, China; ddoapsial@yahoo.com

**Keywords:** endogenous melatonin, exogenous melatonin, growth regulator, bio-stimulator, antioxidants, oxidative stress

## Abstract

Melatonin is a multifunctional signaling molecule, ubiquitously distributed in different parts of plants and responsible for stimulating several physiological responses to adverse environmental conditions. In the current review, we showed that the biosynthesis of melatonin occurred in plants by themselves, and accumulation of melatonin fluctuated sharply by modulating its biosynthesis and metabolic pathways under stress conditions. Melatonin, with its precursors and derivatives, acted as a powerful growth regulator, bio-stimulator, and antioxidant, which delayed leaf senescence, lessened photosynthesis inhibition, and improved redox homeostasis and the antioxidant system through a direct scavenging of reactive oxygen species (ROS) and reactive nitrogen species (RNS) under abiotic and biotic stress conditions. In addition, exogenous melatonin boosted the growth, photosynthetic, and antioxidant activities in plants, confirming their tolerances against drought, unfavorable temperatures, salinity, heavy metals, acid rain, and pathogens. However, future research, together with recent advancements, would support emerging new approaches to adopt strategies in overcoming the effect of hazardous environments on crops and may have potential implications in expanding crop cultivation against harsh conditions. Thus, farming communities and consumers will benefit from elucidating food safety concerns.

## 1. Introduction

Melatonin (*N*-acetyl-5-methoxytryptamine) has an indole ring structure, low molecular weight, and is an evolutionarily conserved pleiotropic molecule that exists ubiquitously in living organisms [1]. It has been recognized that melatonin plays a significant role in animals and plants, particularly in different human processes including generation, sleep regulation, circadian rhythms, retina physiology, sexual behavior, seasonal reproductive physiology, immunological enhancement, and aging [2,3,4,5]. The pleiotropic biological activities of melatonin in living organisms are mediated by membrane receptors and nuclear receptors [6,7]. In addition, melatonin receptors act independently [8], and their bioactive metabolites influence the exchange of melatonin with reactive oxygen species (ROS) and reactive nitrogen species (RNS) [9].

Melatonin has an amphiphilic or amphipathic molecular character. It can easily pass through the cell membrane and dispense to the cytosol, the nucleus, and mitochondria [10]. The amphiphilic character specifies that the membrane receptor is not inevitably essential for facilitating melatonin actions. In fact, melatonin plays an important role in non-receptor-mediated activities such as scavenging ROS and RNS and improving antioxidant capacity, preventing cells, tissues, and organisms from oxidative stress [11,12,13]. Subsequently, the formation and absorption of ROS and/or RNS are elementary process associated with cellular biology and physio-pathology Thus, it is assumed that the primary role of melatonin in living organisms is to improve antioxidant activities and act as a first-line defense against any hazardous conditions [14]. 

Several successive studies have quantified the presence of melatonin in roots, stems, leaves, flowers, fruits, bulbs, and seeds of many plants such as tomatoes, cucumbers, bananas, apples, onions, rice, and so on [15]. It was observed that the content of melatonin enumerated in plant samples differed regularly between cultivars, species, growth and developmental periods, tissue categories, and even in repeats of a single experiment [16]. 

In the last few years, the role of melatonin in plants has been studied progressively. Biotic and abiotic stressors in plants cause growth obstruction, senescence, yield lessening, and even death. The plant develops different physiological activities to lessen the loss induced by means of various stressors. It has been established that melatonin is involved in improving physiological processes, for example, spreading the plant’s normal growth as well as shielding emergent tissues from injury and stress signals from environmental hazards [16,17,18,19,20,21]. In addition, recent reviews have described the significant characteristics of melatonin in plant behavioral responses against environmental stress [4,22]. 

Taking into consideration the new progress in melatonin studies in recent years, the activity of melatonin in plants has been comprehensively and intensely explored. In addition, the mechanisms of action associated with melatonin have progressively been revealed. In the current review, the availability of melatonin in different parts of plants and the biosynthesis pathway of melatonin in plants are summarized. In addition, we focus herein on the growing and developmental parameters, abiotic and biotic stress responses of melatonin in plants, and the mitigation impacts of exogenous melatonin on plant responses to environmental stress features and/or plant–pathogen contacts. Notably, the significance and scope of melatonin research in plants is speculated, which might be supportive and insightful for existing research, and determine the imminent route of melatonin study in plants.

## 2. History of Melatonin in Plants

Melatonin was first discovered in the bovine pineal gland of cows in 1958, and was made known to be the source of melanosome content in vertebrates and fish melanophores [23]. Nowadays, melatonin is one of the widely reviewed natural molecules in living organisms, extending from bacteria to mammals [1,20]. In the first four decades from its discovery, melatonin became well-known as an animal hormone, and research mainly focused on the physiological role of melatonin. Later on, melatonin research focused on dynamic influences including darkness signals [24], transferring communication towards the brain and marginal organs, and acting as a self-coordinator for biological rhythms such as the biological clock and periodic imitation [25]. Research also focused on its palliative properties in many disorders [26], including Alzheimer’s and Parkinson’s syndrome [27], glaucoma, multiple sclerosis, depression, insomnia, chronic fatigue syndrome, schizophrenia, anxiety, metabolic syndrome, osteoporosis, and some forms of cancer [28].

However, in the earlier stages, very few observations on the effects of melatonin in eukaryotic plant cells, such as in endosperm cells of bulbous plants [29] and in epidermal cells of onions [30], were given a new direction of study. Thirty years after the discovery of melatonin in mammals, melatonin was found in the single-celled dinoflagellates and finally transformed its research like the methoxyindole [31]. Although melatonin has been recognized in Japanese morning glories (*Pharbitis nil*) in 1993, its results were not published comprehensively until 1995 [32]. Interestingly, two clear verifications in 1995 pointed to melatonin existence in higher plants [33,34]. In 2003, 108 Chinese medical herbs were selected to determine melatonin content. Melatonin was found in all herbs, with levels ranging from limited nanograms to numerous thousand nanograms per gram of tissue [35]. These huge variances exhibited between the species suggest that the function of melatonin in plants must be varied. After that, several successive studies (Table 1) confirmed the occurrence of melatonin in different plants, even in different parts of individual plant [15,20].

## 3. Biosynthesis of Melatonin in Plants

In general, melatonin can be transformed enzymatically, pseudo-enzymatically, or non- enzymatically into a number of biologically active metabolites such as 5-MT (5-methoxytryptamine), c3OH M (cyclic 3-hydroxymelatonin), AFMK (*N*1-acetyl-*N*2-formyl-5-methoxykynuramine), and AMK (*N*1-acetyl-5-methoxykynuramine) [14,56]. In addition, melatonin degraded to AFMK by numerous enzymes such as IDO (indoleamine 2,3-dioxygenase), EPO (eosinophil peroxidase), HRP (horseradish peroxidase), MPO (myeloperoxidase), CYP (cytochrome P_450_) subforms, and NQR_2_. Furthermore, reference [56] reported the particulars and additional paths of AFMK and AMK formation. Interestingly, CYPs interchangeably break down melatonin to OHM (6-hydroxymelatonin), NAS (*N*-acetylserotonin), or AFMK [14]. The biologically active metabolites of melatonin prominently enlarged the range of this universally acting indoleamine [9,56].

In the case of plants, Murch and Saxena [46] observed that 5-hydroxytryptophan is involved in serotonin synthesis in the flowering plant *Hyericum perforatum* L. (namely Saint John’s wort), similar to vertebrates. However, another study on rice indicated that the conversion of serotonin occurred markedly through tryptophan–tryptamine–serotonin, known as the tryptamine pathway. Subsequently, this pathway has been commonly observed in several plant species [57]. Interestingly, in plants as well in animals serotonin is transformed to NAS, which is further catalyzed by SNAT (serotonin *N*-acetyltransferase) and HIOMT (methylated via hydroxyindole-*O*-methyltransferase), and is recognized as ASMT (acetyl serotonin methyl transferase) and ensures melatonin creation. In addition, *N*-acetyl serotonin is formed in plants from tryptamine, where *N*-acetyltryptamine assists as a transitional product, and is assembled by HIOMT/ASMT and SNAT [58,59]. In the meantime, indole acetic acid is formed from tryptamine, and indole-3-acetylaldehyde acts equally as a middle product [58].

In brief, as shown in Figure 1, the standard pathway of melatonin biosynthesis from tryptophan in plants involves four steps: first, TDC (decarboxylation by tryptophan decarboxylase); second, T5H (amine hydroxylation by tryptamine 5-hydroxylase) to serotonin; third, *N*-acetylation through SNAT (serotonin *N*-acetyltransferase), which activates the similar reaction AANAT (non-homologous aralkylamine *N*-acetyltransferase) of vertebrates; and finally, the *O*-methylation to melatonin via ASMT (*N*-acetylserotonin *O*-methyltransferase) [60,61].

## 4. Melatonin Acts as a Plant Growth Regulator

The chemical compounds that are synthesized artificially and act as plant hormones in regulating the growth of cultivated plants, weeds, and in-vitro grown plants and plant cells are usually known as plant growth regulators. In recent years, numerous studies have promoted that melatonin can be one of the core actors in the response mechanism and might have significant capabilities in plant physiology under adverse environments. The concentrations of melatonin were differed extensively in different plant species, even in different plant parts of same species. This undeniably concludes that melatonin must have diverse functions by means of growth regulators in plants [20]. The auxin plays a diverse role in plant growth and development, and Indole-3-acetic acid (IAA) is considered a most common auxin. Structurally, melatonin and IAA showed many similarities such as a planar aromatic ring, a carboxylic acid-binding site, and a hydrophobic transition region. Evidently, both melatonin and IAA could partially contribute to varied physiological processes relating to plant growth and development, and antioxidant potentials as well [63]. Melatonin acts as a growth developer in *Lupinus albus* (as like IAA), and encourages the vigorous growth of hypocotyls at micromolar accumulations, even though it has inhibitory consequences at higher accumulations [58]. Compared to IAA, the impact of melatonin on growth promotion is 63% greater, which is considered a significant auxinic outcome [50]. Similarly, the assessment of melatonin on growth promotion in other plants, such as numerous monocots, oats, wheat, barley, and canary grasses, ranged from 10% to 55% relative to IAA. Moreover, similar to IAA, the growth restrictive effects of melatonin on the roots of canary grass and wheat were about 56% and 86%, respectively, compared to IAA [42]. The authors [64] in Brassica observed that the lower accumulation of melatonin prompted the IAA biosynthesis, which resulted in the encouragement of root growth, though the particular association between IAA and melatonin is not clear yet. In addition, the melatonin effect on the stimulation of rhizogenesis was primarily confirmed in 2007. The induction of adventitious or lateral roots in lupin, prompted through the consequence of melatonin, persuaded root primordials from pericycle cells. Recently, it has been confirmed that the rhizogenic effect was noticeable in cucumbers, cherry rootstocks, rice, and pomegranates [58]. 

Additionally, melatonin had a significant influence as a plant growth regulator in cell culture. The endogenous level of melatonin in the culture medium of in-vitro culture explants of *Hypericum perforatum* modified the plant morphogenesis, altering auxin-induced rhizogenesis and cytokinin-induced caulogenesis, indicating that melatonin had a potential impact on plant growth regulation and auxin modulation [65,66]. Moreover, it has been observed that the rice IDO (indoleamine 2,3-dioxygenase) gene showed over expression in transgenic tomatoes, indicating auxin-like action of melatonin in apical dominance and branching [67]. Interestingly, low levels of biosynthetic melatonin over expressed the OID gene in transgenic tomato plants, leading to lateral leaflet pattern changes, a drop in leaflet number, as well as less firm and more serrated leaves compared to wild type. However, over expression of SNAT and HIOMT in tomato plants attributed higher melatonin levels in leaves, and led to a considerable decline in endogenous IAA levels because of tryptophan, a common ancestor of melatonin and IAA [68]. 

Although the numerous functions of melatonin have been scrutinized thoroughly in higher plants, the data are limited in its altogether role. In summary, several studies recommended exact physiological actions of melatonin in plants, such as stimulating the growth of different seedlings [42,69], triggering or preventing the growth of primary roots [69,70], encouraging lateral and adventitious rooting in different species [70,71], adjusting branching and growth patterns of stems and leaves [68], inhibiting delay-induced leaf senescence through enhancing photosynthesis, CO_2_ uptake, and biomass accumulation [57,72], stimulating rhizogenesis and caulogenesis in explant cultures [66,73], effects on flowering [74], and altered levels during fruit development and seed formation [54,75]. 

## 5. Melatonin Acts as a Bio-Stimulator and an Antioxidant in Plants

Plants are sessile, but they can modify their own physiological condition to adjust to harsh environmental conditions. In any plant challenged to an unfavorable environment, a quick and remarkable variation arises inside the plant cells to stay alive. Thus, different bio-stimulators are activated in the adjustment to the harsh environment to encourage the prevailing capabilities of bioremediation. Likewise, melatonin can boost the physiological activity against adverse environments as an efficient antioxidant compound. More than 25 years ago, it was found that melatonin acts as an uninterrupted free radical scavenger [76]. Melatonin is one type of amphiphilic indole ring structure compound, and it moves without difficulty via cell membranes to the cytoplasm. It can also pass subcellular partitions due to its amphipathic indole ring structure. For these reasons it is called an ecologically friendly molecule, having wide ranges in antioxidant capacity [77,78]. Normally, a cell is isolated from its adjacent surroundings by the plasma membrane. This physical barrier (plasma membrane) is particularly absorbent to small molecules (even to ions), but melatonin crosses the barrier easily due to its amphipathic nature [62]. Similarly, cyclic 3-hydroxymelatonin has antioxidant properties that are capable of counteracting the extremely toxic hydroxyl radical (OH•). The reaction of melatonin with hydrogen peroxide was shown in vitro in 2000 by Tan et al., and it was reported that its product also had antioxidant properties [79]. Melatonin is also known to scavenge the superoxide (O^2−^) [80]. These scavenging activities occurred due to the surprisingly extraordinary competence of melatonin in dropping radical loss in vivo [81]. Poeggeler et al. [82] stated that melatonin is five times more effective than glutathione (GSH) in neutralizing hydroxide (OH^-^) and 15-fold more effective than the exogenous scavenger mannitol. 

Besides the role of melatonin in directly scavenging numerous free radicals, ROS, and RNS, it also acts as a signaling molecule at the cellular level and up-regulates a number of antioxidant enzymes, which increases its efficiency as an antioxidant [83]. The interaction between melatonin and ROS in plants indicates the function of melatonin as an effective antioxidant through both direct and indirect mechanisms. Melatonin signal transmission acts on ROS-mediated signals such as the balancing of hydrogen peroxide (H_2_O_2_). Melatonin acts as a direct antioxidant and is proficient in lowering the levels of reactive oxygen compared to ascorbic acid. Various melatonin metabolites, such as 3-OHM, AFMK, and 2-hyxdroxymelatonin, also act by means of influential antioxidants, promoting the antioxidant capabilities of this biomolecule [84]. Remarkably, melatonin is not limited by any prerequisite for an exact recovering pathway or any supplementary metabolites for accomplishment of a redox cycle [85,86]. Melatonin acts as a mediator in different antioxidant pathways, for example, the glutathione ascorbate cycle, peroxidases, superoxide dismutase, and catalase through varied mechanisms, resulting in abiotic and biotic stress responses in the plant [22,87]. Moreover, reactive nitrogen species, for example, nitric oxide, are also detoxified by melatonin [88]. 

It was hypothesized that the pathway formed with melatonin can play a significant role in reclaiming ROS and RNS intended for comeback from growing atmospheric oxygenation [4,60]. The melatonin and ROS interface acts as a vital and fast signaling molecule within plants when ROS arises [89,90]. Therefore, based on several findings, it can be summarized that there is a dynamic association between melatonin and electrical, ionic, and chemical signaling pathways, which results in adaptive activities and a transformed plant metabolism for developing tolerance to hazardous environments (Figure 2).

## 6. Exogenous Melatonin and Plant Stress Tolerance

Recently, the impact of exogenous melatonin on plant defense systems to cope with biotic and abiotic stress has attracted the attention of the researchers worldwide. Although the pathways of melatonin synthesis in plants and its metabolic mechanisms are still unclear, researchers still believe that melatonin production in plants is a universal fact. Relatively greater levels of melatonin in plants results in production of RNS and ROS, which in turn, plays an important role in plant tolerance against adverse environmental conditions [91]. The detoxification system in plants fails to completely degrade some foreign compounds, which leads to the increased level of toxicity in plants [92]. Melatonin developed a new era in plant adaptation mechanisms. Therefore, studying exogenous melatonin in stressed plants, in terms of plant adaptation and survival, has been gaining extensive consideration in researcher. Recently, it has been confirmed that exogenous melatonin improves the tolerance against salt, drought, and cold stress in Bermuda grass [93]. In some plant species, the impact of exogenous melatonin, regarding stress management, is dose dependent [62]. In *Brassica juncea*, Chen et al. [64] demonstrated that root growth was stimulated in response to low concentrations of melatonin (0.1 mM), whereas a high concentration (100 mM) inhibited root growth. Similar results were also observed in cherry tissue culture [71]. They demonstrated that melatonin administration at high concentrations declined in total biomass content, which could be explained by the influence that melatonin at high concentrations has in decreasing the endogenous level of promoters. A number of authors showed that melatonin application substantially alleviated abiotic and biotic stress responsive reduction in growth, biomass accumulation, chlorophyll loss, photosynthetic inhibition, and antioxidant activities in several plants, as described in Table 2. It has been demonstrated that glutathione peroxidase (GPX), glutathione reductase (GR), Zn, Cu and/or Mn-SOD, peroxidase (POD), and catalase (CAT) activities were up-regulated as free radical scavengers to reduce ROS in plant cells through exogenous melatonin. They also reported that exogenous melatonin not only protected against ROS, but also protected the proteins related to chlorophyll and photosynthesis. They further mentioned that the increase in efficiency was due to the better working of photosystem II, having a larger number of open reaction centers that are better for the function of all the photosynthetic transport chain members under melatonin treatment. Remarkably, exogenous melatonin prevented the loss of chlorophylls, and it could be explained by the stimulation of Fe uptake and the subsequent increase in ferrodoxins, which regulate the amount of reduced ascorbate and protect chlorophyll from degradation [94]. They also reported that melatonin results in suppression of salt-induced inhibition and enhancement of the ferrodoxins gene *PetF* in soybeans. Exogenous melatonin also protects cell ultrastructure. Recently, reference [95] demonstrated that seed priming with melatonin produced seedlings with a 20% increase in root length and improved and organized cell ultrastructure. Similarly, recovery of leaf ultrastructures through exogenous melatonin have been observed in tomatoes that were stressed in acid rain conditions [17]. 

In addition, melatonin transgenic plants have transformed, or act together with, other phytohormones to further adjust different growing processes in hostile environments. For example, application of melatonin boosted the root growth and development of transgenic rice plants [70]. Moreover, it was observed that rice *ASMT*-mRNA was markedly expressed after abscisic acid and methyl jasmonic acid treatments, specifying the potential involvement of melatonin under different stress conditions [96]. Therefore, the evidence from different observations suggested that transgenic plants conveying melatonin biosynthesis genes have generated physiological activities and improved stress tolerance abilities of plants under unfriendly conditions [97,98].

Moreover, exogenous melatonin modulated the expression of numerous genes in plants [62,99]. Interestingly, it was also observed that the genes controlled by low melatonin concentrations might not be controlled by high melatonin concentrations [99]. Genome-wide transcriptomic profiling revealed that genes were differentially expressed in melatonin-treated plants, compared to controls. Gene ontology enrichment studies characterized various genes related to different primary and secondary plant physiological metabolisms (nitrogen metabolism, carbohydrate metabolism, tri-carboxylic acid transformation, transport, hormone metabolism, metal handling, and redox) in melatonin pre-treated plants. Melatonin plays a pivotal role in the regulation of several specific stress related genes. For example, chlorophyll content may be preserved via melatonin, and the light-regulating enzyme associated with chlorophyll degradation, namely chlorophyllase (CLH1), was significantly down-regulated with melatonin treatment in Arabidopsis [99]. Other research also reported that inhibited pheide-a-oxygenase (PAO) transcript levels were inhibited via exogenous melatonin. PAO is another vital enzyme that degrades chlorophyll [100]. In agreement with these findings, Zhang et al. [62] confirmed that exogenous melatonin plays a significant role in protecting chlorophyll content in leaves and also helps in delaying the senescence and boosting photosynthetic rates. Therefore, melatonin, in conjunction with antioxidant enzymes, improves photosynthesis, delays leaf senescence, slows alterations in the leaf ultrastructure, delays biosynthesis of metabolites, and modulates stress genes to form an efficient system that protects plants from harsh environments. All these outcomes suggest that melatonin has an imperative role in coping with harsh environmental conditions.

## 7. Conclusions

Melatonin is a pleiotropic molecule and has amphiphilic properties in plants. The current review discussed the properties of melatonin, and the recent progress in research. Melatonin presents in different parts of the plant, and melatonin biosynthesis leads to the development of crucial functions in plants for surviving against various stresses. Melatonin played an important role in mitigating abiotic and biotic stress directly through scavenging ROS and RNS, and indirectly through recovering leaf ultrastructure, improving the photosynthesis system, stimulating plant growth regulators, and triggering antioxidant activities in plants. Nonetheless, endogenous melatonin, in addition to other natural protectors in plants, is unable to protect plants against severe stress conditions. In this regard, exogenous melatonin showed remarkable coping mechanisms in harsh environments by boosting plant growth regulation, delaying leaf senescence, increasing photosynthesis, and increasing ROS and RNS scavenging antioxidant systems in plants. Meanwhile, the physiological and molecular activities of melatonin in plants indicate that melatonin is an essential molecule in the stimulation of field crops, especially where biotic and abiotic stress is a limiting factor for crop production.

However, there are numerous major issues to be explored. The role of endogenous melatonin and the uses of exogenous melatonin against viruses, nematodes, or insects requires detailed investigations. There is still lack of information available regarding the genes and core pathways that are precisely regulated by melatonin. To conclude, there is enormous research potential for bettering our understanding of the impact that melatonin has in basic life functions across plant kingdoms, and the creation of new approaches to advance progress in plant cultivation and industrial agriculture. 

## Figures and Tables

**Figure 1 ijms-20-01040-f001:**
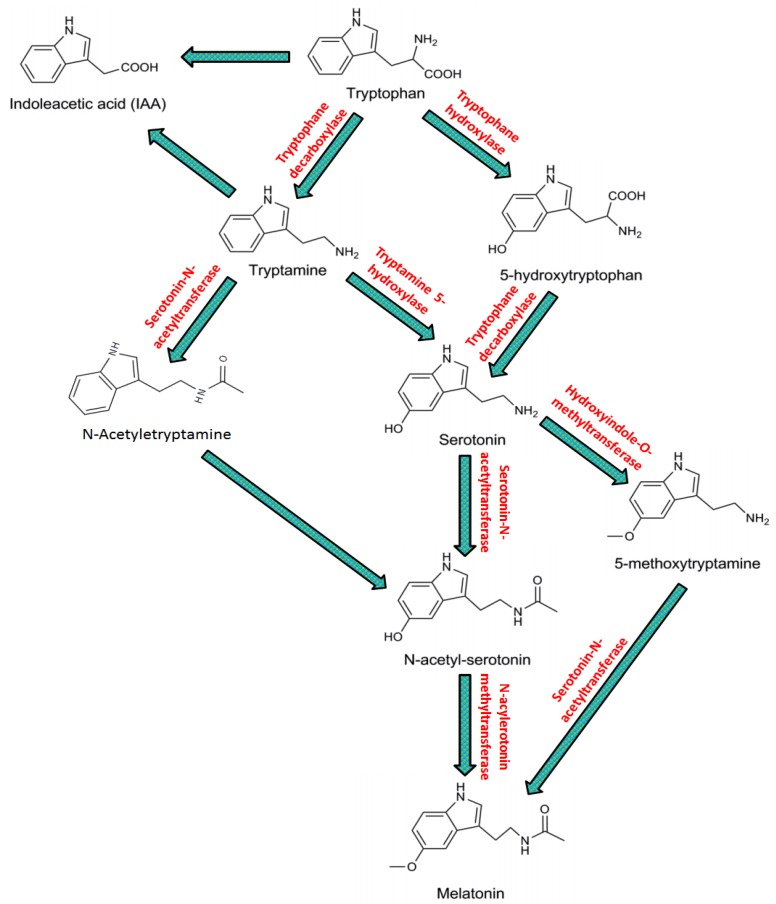
Biosynthesis of melatonin in plants. Modified from Zhang et al. [62].

**Figure 2 ijms-20-01040-f002:**
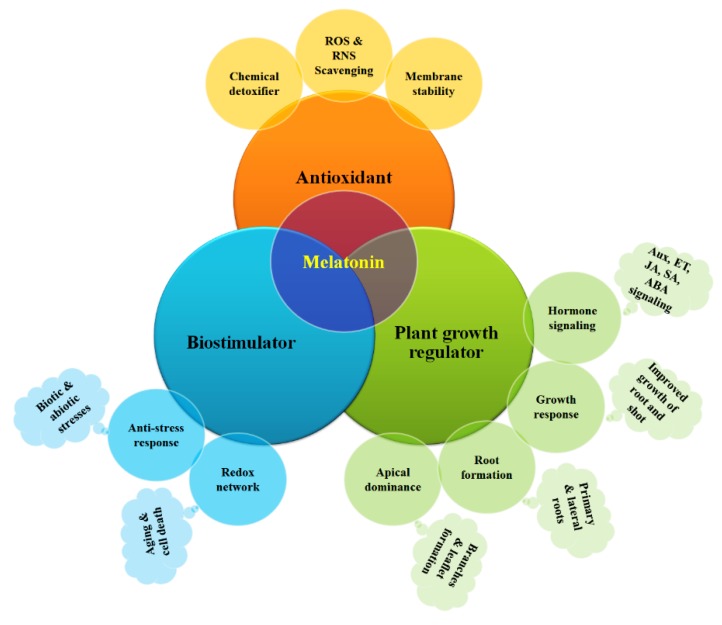
Mechanism of melatonin activities in plants as a growth regulator, bio-stimulator, and antioxidant. Here, ROS, RNS, Aux, ET, JA, SA, ABA indicates reactive oxygen species, reactive nitrogen species, auxin, ethylene, jasmonic acid, salicylic acid, and abscisic acid, respectively.

**Table 1 ijms-20-01040-t001:** Availability of melatonin in different plant parts.

Plant Name	Family	Plant Parts								Reference
		Seed	Leaf	Shoot	Flower	Fruit	Coleoptile	Root	Bulb	
Alfalfa	Fabaceae									[36]
Almond	Rosaceae									[37]
Aloevera	Asphodelaceae									[35]
Anise	Apiaceae									[38]
Apple	Rosaceae									[39]
Arabidopsis	Brassicaceae									[40]
Banana	Musaceae									[41]
Barley	Poaceae									[42]
Beet	Amaranthaceae									[33]
Broccoli	Brassicaceae									[36]
Cabbage	Brassicaceae									[36]
Canary grass	Poaceae									[42]
Carrot	Apiaceae									[39]
Celery	Apiaceae									[38]
Cherry	Rosaceae									[43]
Chilies	Solanaceae									[44]
Chinese liquorice	Fabaceae									[39]
Coriander	Apiaceae									[45]
Corn	Poaceae									[45]
Cucumber	Cucurbitaceae									[39]
Fennel	Apiaceae									[38]
Fenugreek	Fabaceae									[38]
Fever few	Asteraceae									[46]
Garlic	Amaryllidaceae									[39]
Grape	Vitaceae									[47]
Green cardamom	Zingiberaceae									[38]
Ginger	Zingiberaceae									[39]
Huang-qin	Lamiaceae									[37]
Kiwifruit	Actinidiaceae									[34]
Lupin	Fabaceae									[41]
Maize	Poaceae									[39]
Mango	Anacardiaceae									[48]
Milk thistle	Asteraceae									[38]
Morning glory	Convolvulaceae									[49]
Mung bean	Fabaceae									[36]
Mustard	Brassicaceae									[38]
Oat	Poaceae									[42]
Onion	Amaryllidaceae									[36]
Orange	Rutaceae									[48]
Papaya	Caricaceae									[48]
Pineapple	Bromeliaceae									[39]
Pomegranate	Lythraceae									[39]
Poppy	Papaveraceae									[38]
Potato	Solanaceae									[50]
Radish	Brassicaceae									[36]
Red pigweed	Chenopodiaceae									[51]
Rice	Poaceae									[45]
Sage	Lamiaceae									[52]
St. John’s wort	Hypericaceae									[46]
Strawberry	Rosaceae									[53]
Sunflower	Asteraceae									[38]
Tall fescue	Poaceae									[34]
Tobacco	Solanaceae									[33]
Tomato	Solanaceae									[54]
Turnip	Brassicaceae									[39]
Walnut	Juglandaceae									[55]
Wheat	Poaceae									[42]

**Table 2 ijms-20-01040-t002:** Exogenous melatonin improves abiotic and biotic stress tolerance in different plants.

Crops Name (Scientific Name)	Family	Different Stress Tolerances
Arabidopsis (*Arabidopsis thaliana*)	Brassicaceae	Salinity [101]; Drought [98]; High temperature [83]; Low temperature [102]; Pathogen [88]
Rapeseed (*Brassica napus* L.)	Brassicaceae	Drought [103]
Tomato (*Solanum lycopersicum*)	Solanaceae	Salinity [104]; Sodic Alkaline [105]; Drought [106]; High temperature [107]; Low temperature [108]; Heavy metal [109]; Acid rain [17]
Potato (*Solanum tuberosum*)	Solanaceae	Pathogen [110]
Cucumber (*Cucumis sativus*)	Cucurbitaceae	Salinity [111]; Nitrate [112]; Drought [113]; Low temperature [114]
Watermelon (*Citrullus lanatus*)	Cucurbitaceae	Salinity [19]; Low temperature [115]; Heavy metal [116]
Rice (*Oryza sativa*)	Poaceae	Low temperature [117]; Low temperature & drought [118]
Wheat (*Triticum aestivum*)	Poaceae	Low temperature [119]; Drought [18]; Heavy metal [120]
Maize (*Zea mays*)	Poaceae	Salinity [121]; Drought [122]
Ray grass (*Lolium perenne*)	Poaceae	High temperature [123]
Tall fescue grass (*Festuca arundinacea*)	Poaceae	High temperature [124]
Naked oat (*Avena nuda* L.)	Poaceae	Drought [125]
Alfalfa (*Medicago sativa*)	Fabaceae	Drought [126]; Heavy metal [127];
Sunflower (*Helianthus annuus*)	Asteraceae	Salinity [128]
Crabapple (*Malus hupehensis*)	Rosaceae	Salinity [129]; Alkaline [130]
Peach (*Prunus persica*)	Rosaceae	Low temperature [131]
Banana (*Musa acuminate*)	Musaceae	Pathogen [132]
Kiwifruit (*Actinidia deliciosa*)	Actinidiaceae	High temperature [133]
Tea (*Camellia sinensis* *L.*)	Theaceae	Low temperature [134]

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
