# Peer review of "Melatonin Mediates Enhancement of Stress Tolerance in Plants"

_ijms, 2019, doi:10.3390/ijms20051040_

Round 1

Reviewer 1 Report

I carefully read the review. Currently, it is not easy to write a brilliant review of the role of melatonin in plants. The reason is that annually in the leading biological journals published very thorough reviews on this topic. The authors of this review in a brief form consider a wide range of problems on the protective role of melatonin in plants. One of the advantages of the review is its brevity and great informational content. I believe that the review as a whole is interesting and useful, although it does not provide fundamentally new generalizations from the available data. Unfortunately, the authors did not try to consider all the material available in the literature from some new point of view. They only summarized the literary data, but they did it quite successfully.

It seems to me that for the first time in this review, the negative effects of high concentrations of melatonin are analyzed. Until now, the authors of other papers have described exclusively the positive effects of melatonin. In contrast to other reviews, in this case data on the endogenous melatonin content are presented not only in different plant species, but also in different organs of the same plant. The paper presents an attempt to characterize melatonin as a plant growth regulator, a bio-stimulator and an antioxidant. At the same time, the authors do not give a precise definition of the terms plant growth regulator and bio-stimulator. This creates an uncertainty in the reader when analyzing the presented literature data. In addition, comparing the effect of melatonin on growth processes with the influence of auxins, the authors absolutely do not discuss the possibility of involving auxin in the realization of the biological action of melatonin. In addition, it seems to me that more active involvement of data obtained on transgenic plant forms and mutant lines would make some of the points of this work more evidentiary.

In general, I believe that the review may be published after some refinement.

Author Response

Author’s Response to Reviewer-1 Comments

Manuscript ID:ijms-435230

Manuscript Title: Melatonin Mediates Enhancement of Stress Tolerance in Plants

The authors would like to thank the Editor and Reviewer-1 for their deep review of our manuscript with very constructive comments and suggestions. The general and specific questions or concerns and recommendations are highly appreciated, and corrections have been made accordingly in the revised version of the manuscript to improve its quality. The point-by-point responses to the Reviewer-1 comments and suggestions are listed as follows:

Reviewer #1 Comments: 
I carefully read the review. Currently, it is not easy to write a brilliant review of the role of melatonin in plants. The reason is that annually in the leading biological journals published very thorough reviews on this topic. The authors of this review in a brief form consider a wide range of problems on the protective role of melatonin in plants. One of the advantages of the review is its brevity and great informational content. I believe that the review as a whole is interesting and useful, although it does not provide fundamentally new generalizations from the available data. Unfortunately, the authors did not try to consider all the material available in the literature from some new point of view. They only summarized the literary data, but they did it quit successfully.

It seems to me that for the first time in this review, the negative effects of high concentrations of melatonin are analyzed. Until now, the authors of other papers have described exclusively the positive effects of melatonin. In contrast to other reviews, in this case data on the endogenous melatonin content are presented not only in different plant species, but also in different organs of the same plant. The paper presents an attempt to characterize melatonin as a plant growth regulator, a bio-stimulator and an antioxidant. At the same time, the authors do not give a precise definition of the terms plant growth regulator and bio-stimulator. This creates an uncertainty in the reader when analyzing the presented literature data. In addition, comparing the effect of melatonin on growth processes with the influence of auxins, the authors absolutely do not discuss the possibility of involving auxin in the realization of the biological action of melatonin. In addition, it seems to me that more active involvement of data obtained on transgenic plant forms and mutant lines would make some of the points of this work more evidentiary.

In general, I believe that the review may be published after some refinement.

Response: Thank you so much for your deep review and kind consideration. We have added definition of plant growth regulator and bio-stimulator according to your kind suggestion. Please see the line 126-128 and 176-178. We have added the influence of auxin as you suggested. Please see the line 133-148. In addition, we have added some information on transgenic plants form according to your kind advice. Please see the line 251-258.

Reviewer 2 Report

The manuscript does not present relevant errors, it is well written and understood very well. The review is sufficiently complete. It only presents few typographical errors easily remedied by the edition.

Author Response

Author’s Response to Reviewer-2 Comments

Manuscript ID:ijms-435230

Manuscript Title: Melatonin Mediates Enhancement of Stress Tolerance in Plants

The authors would like to thank the Editor and Reviewer-2 for their deep review of our manuscript with very constructive comments and suggestions. The general and specific questions or concerns and recommendations are highly appreciated, and corrections have been made accordingly in the revised version of the manuscript to improve its quality. The responses to the reviewer-2 comments are as follows:

Reviewer #2 Comments: 

The manuscript does not present relevant errors, it is well written and understood very well. The review is sufficiently complete. It only presents few typographical errors easily remedied by the edition.

Response: Thank you so much for your kind deep review and kind consideration.
